# Using the Ratio of Urine Testosterone to Estrone-3-Glucuronide to Identify the Sex of Chinese Giant Salamanders (*Andrias davidianus*)

**DOI:** 10.3390/ani12091112

**Published:** 2022-04-26

**Authors:** Jianlu Zhang, Jiqin Huang, Hu Zhao, Jie Deng, Fei Kong, Hongxing Zhang, Qijun Wang

**Affiliations:** 1Shaanxi Key Laboratory for Animal Conservation, Shaanxi Institute of Zoology, Xi’an 710032, China; zhangjianlu@ms.xab.ac.cn (J.Z.); huangjq1985@163.com (J.H.); zhaohu2007@126.com (H.Z.); dengjie0311@ms.xab.ac.cn (J.D.); k.coffee@163.com (F.K.); zhs@ms.xab.ac.cn (H.Z.); 2Key Lab of Agricultural Animal Genetics, Breeding and Reproduction of Ministry of Education, College of Fisheries, Huazhong Agricultural University, Wuhan 430070, China

**Keywords:** Chinese giant salamander (*Andrias davidianus*), urine, testosterone, estrone-3-glucuronide, sex identification

## Abstract

**Simple Summary:**

Due to environmental pollution, habitat loss, and overutilization, the wild population of Chinese giant salamanders (*Andrias davidianus*) is decreasing continuously. Therefore, artificial proliferation and release are effective ways to restore the wild population of this species. However, the sex of Chinese giant salamanders (CGSs) needs to be confirmed before reintroduction or rejuvenation into the wild. The sex identification of amphibians is difficult and is usually conducted through blood hormone testing, ultrasound, or even the observation of gonads and other invasive sampling methods, and the operation is cumbersome and expensive. As one of the main aspects of animal welfare, non-invasive or minimally invasive sampling is increasingly important in wildlife research. As one of the raw materials of non-invasive sampling, urine has been considered by more and more researchers engaged in wildlife physiology and biochemistry. Our objective was to identify the sex of CGSs by detecting the ratio of hormone testosterone (T) to estrone-3-glucuronide (E1G) in urine collected by minimally invasive sampling. Perhaps this method can be used for the determination of sex in other animals, especially amphibians.

**Abstract:**

Minimally invasive sampling was used to determine the sex of Chinese giant salamanders (*Andrias davidianus*). Urine samples (*n* = 25) were collected from 6 adults in the breeding season and from 19 individuals (7 adults and 12 juveniles) in the non-breeding season. The hormone testosterone (T) and estrone-3-glucuronide (E1G) in urine were collected from Chinese giant salamanders (CGSs), and the hormone extracts were analyzed by enzyme immunoassays (EIA). The data demonstrated that the urine T concentration of the male CGSs was significantly higher than that of the females during the breeding season (*p* < 0.05) and even more pronounced during the non-breeding season (*p* < 0.01). The urine E1G concentration of the males was less pronounced than that of the females during the breeding season (*p* < 0.01) and significantly lower during the non-breeding season (*p* < 0.05). The urine T/E1G values of all the male salamanders were significantly higher than those of the females (*p* < 0.01) during both the breeding season and the non-breeding season. An interesting pattern was found in this study: the value of urine log_10_(T/E1G) of the male CGSs was higher than 1, whereas the value for the females was lower than 1, during both the breeding and non-breeding seasons, and in the adult and sub-adult age groups of CGSs. There were 25 salamanders in this study and the accuracy rate reached 100% by using a log_10_(T/E1G) value of 1. The results of the log_10_(T/E1G) value provide new insight into the future development of the sex identification of CGSs and also lay the foundation for accurate sex identification in the preparation for artificial release. This is the first study to show that the T/E1G ratio in urinary hormones is reliable for the sex identification of CGSs. Additionally, urinary hormone T/E1G measures are promising sex identification tools for amphibian or monomorphic species and for those whose secondary sex characteristics are visible only during the breeding season.

## 1. Introduction

The Cryptobranchid (giant salamanders) family consists of two extant genera and three currently recognized species, including the Japanese giant salamander (*Andrias japonicus*) and the North American giant salamander (*Cryptobranchus alleganiensis*) [1,2]. The Chinese giant salamander (*A. davidianus*) is endemic to China and is the largest amphibian of the three extant species of the family *Cryptobranchidae* (Figure 1). Moreover, the family *Cryptobranchidae* is distantly related to the family Salamandridae that emerged some 350 million years ago. As such, the Chinese giant salamander (CGS) is also considered a living fossil [3,4,5] since it represents a transitional form between aquatic and terrestrial organisms and is considered a valuable model for studying vertebrate evolution and biodiversity [3,6]. Due to the low number of wild resources caused by environmental change, water pollution, habitat loss, and overutilization in the past 50 years, the CGS has been designated as part of the national class ΙΙ protected species in China and listed in Appendices I of the Convention on International Trade in Endangered Species of Wild Fauna and Flora (CITES, 2014) since 1975. This species is also categorized as critically endangered by the IUCN Red List [7,8,9,10,11,12,13,14,15,16] because of its critically endangered status and apparent lack of recovery in the wild [14]. Moreover, the decline in the extant populations of the giant salamander in first- to third-order streams could have profound impacts on the trophic ecology within these aquatic systems [15]. 

The farming of the CGS has been a great success in China since 2005, and approximately, 2 million CGSs are artificially reproduced per year [10]. In order to restore the historical range of the species, the rejuvenation and reintroduction of the population, funded by the Chinese government, is used as a conservation tool for the CGS via creating a captive breeding population, which has strict requirements for the sex ratio [17]. A reasonable sex ratio (1:1) is the guarantee of an artificial, successful, and stable population balance of CGSs. Unfortunately, sex identification is difficult for CGSs, especially for juveniles. Accurate sex identification has always been the “bottleneck” problem for the selection and breeding of parent CGSs in the development of the CGS breeding industry [10,12]. Moreover, sex identification is also needed in the commercial breeding farms for juvenile giant salamanders.

In the breeding season, experienced breeders can identify the sex of CGSs by their behavior, abdominal skin folds, body size, and cloacal orifice. However, there are some disadvantages to the above methods because the giant salamander is late in sexual maturity, and these characteristics are illegible in the non-breeding season. Moreover, the sex of salamanders can be determined by directly scanning the salamander’s gonads with ultrasonography methodologies or by drawing blood from the salamander’s tail vein, in which the levels of serum hormones such as estradiol, progesterone, and testosterone (T) can be detected. However, ultrasonography is more expensive and only suitable for adult CGSs during breeding season, and drawing blood is invasive to the CGS [18,19].

As one of the main components of animal welfare, non-invasive hormone monitoring has become an important part of wildlife and conservation research, especially in mammals and birds [20,21,22,23,24]. These techniques, which measure hormone metabolites in voided urine or feces, allow reproductive monitoring, stress evaluation, and sex identification in animals, either from a distance or with minimal handling. For the most part, this work has been carried out in mammals, with a few studies on birds and reptiles. In recent years, non-invasive hormone monitoring has been applied to amphibians in several sexually dimorphic anuran species [25,26,27,28].

At present, most studies on biological estrogens focus on estradiol, but there are no significant sex differences in estradiol concentration in some species. Therefore, researchers also began to select E1G, the primary metabolite of estradiol, as an indicator for detection. Although no relevant experiments have been conducted on amphibians, E1G has been used as a detection indicator for mammals. For example, E1G has been used as a major detection indicator in the study on the reproductive hormone pattern of female gibbons (*Hoolock Leuconedys*) during the whole maturity stage [29]. Moreover, a significant relationship was found between the color and mean concentration of the fecal E1G of white-cheeked gibbons (*Nomascus leucogenys*) [30]. Changes in E1G in feces have also been used as a method to detect estrus in sea lions (*Eumetopias Jubatus*) [31].

Previous studies found that there were no significant differences in the expression of estradiol in the blood of male and female CGSs [32]; hence, urine T and E1G, which are both primary metabolites of estradiol, were selected as research indicators of CGSs in this study. The aim of our study was to find a new minimally invasive method for the rapid and accurate sex identification of CGSs, which will benefit the captive breeding and the design of future rejuvenation and reintroduction programs for this critically endangered species.

## 2. Materials and Methods

### 2.1. Study Animals

On 27 May 2017, the urine of 6 sexually mature CGSs (6 years old, 3 males and 3 females) from a giant salamander farm in the Chenggu county of Shaanxi Province was collected and stored at −80 °C for future use. On 11 November 2021, 19 CGSs were purchased from another giant salamander farm in Xi’an city, at the northern foot of the Qinling Mountains. Among them, 7 salamanders were sexually mature (7.5 years old, 3 males and 4 females) and 12 were sub-adults (4.5 years old). The salamanders were deep anesthetized in a water bath containing tricaine methane sulfonate (MS-222) at a concentration of 600 mg·L^−1^. The method of anesthesia was based on the results of our previous study [33], and the anesthesia lasted about 35 min for the 7.5-year-old CGSs and 30 min for the 4.5-year-old CGSs. Finally, the deep anesthetized CGSs were euthanized for dissection to confirm their sex after the urine was collected. The urine was stored in an ultra-low-temperature freezer (−80 °C) until analysis for T and E1G hormones.

### 2.2. Husbandry, Management, and Sexing of the Animals

The experimental salamanders were placed in white foam boxes with a size of 60 × 40 × 20 cm and a water depth of 8 cm when purchased from aquafarms, then temporarily bred in the laboratory for two weeks and fed with fresh wild fish at 9 a.m. every day before urine collection. The sex of the CGSs could be determined by the morphological characteristics of the gonads. The female gonads showed a wide band of milky white, and the middle and rear sections were obviously curved, showing an “S” line. The eggs on the surface of the ovary could be clearly observed visually. 

### 2.3. Experimental Reagents and Consumables

A Testosterone Enzyme Immunoassay Kit (Arbor Assays, catalog number k032-H1), an Estrone-3-Glucuronide Enzyme Immunoassay Kit (Arbor Assays, catalog number k036-H1), and a 15 mL centrifuge tube (for collecting urine samples), etc., were used in this experiment. 

### 2.4. Experimental Apparatus

An enzyme standard instrument (Bio-Rad, Hercules, CA, USA) was used as the experimental apparatus in this study. 

### 2.5. Experimental Methods

The concentrations of T and E1G in the urine of 6 sexually mature CGSs during breeding season and 19 salamanders during non-breeding season (7 sexually mature adults and 12 sub-adults) were detected by ELISA. The specific detection and analysis methods are described in the following section.

#### 2.5.1. Urine Collection

After binding a CGS with a cloth bag, the CGS was placed on its back on the experimental table. The water around the cloacal orifice was gently wiped off with a clean dry towel, and urine was collected by gently squeezing the salamander’s abdomen. A 50 mL centrifuge tube was laced in the front of the orifice of the salamander to collect urine and was then quickly stored in a refrigerator at −80 °C for future detection.

#### 2.5.2. Steps for Determining the Concentration of T and E1G in Urine

We made a standard curve according to the instructions of the Testosterone Enzyme Immunoassay Kit and the Estrone-3-Glucuronide Enzyme Immunoassay Kit, respectively. The T and E1G concentrations were detected using the two kits, following the manufacturer’s protocol.

### 2.6. Statistical Analysis

The standard curve was drawn by the Hill fitting curve method in ELISACalc. An independent sample T test was performed for the concentration of T and E1G in urine, obtaining the log_10_(T/E1G) of male and female CGSs using SPSS Statistics 24.0 software. GraphPad Prism 8.0.2 was used to plot the results, and the significant difference (*p* < 0.05) and extremely significant difference (*p* < 0.01) were represented by * and **, respectively. The T tests plus the paired test were used for the data statistics. 

## 3. Results

### 3.1. Determining the Concentration of T and E1G in the Urine of CGSs during the Breeding Season

EIA was used to determine the T and E1G concentration’s standard curve. The urine of six CGSs (three males, three females) was collected for three-fold dilution. Then, the T and E1G concentration was detected. The T and E1G concentration in the urine of the six salamanders is shown in Table 1. Hill curve fitting was adopted, and the standard curve equations of the two hormones were y_T_ = 243.00903/(103.8438 + x^0.66824, ***R*^2^** = 0.9961), and y_E1G_ = 386.21935/(240.21305 + x^1.18952, ***R*^2^** = 0.9854). The T concentration in the urine of the males (4289.085 ± 2135.082) was significantly higher than that of the female salamanders during the breeding season (67.971 ± 12.406), (*p* < 0.05, Figure 2). The concentration of E1G in the males (12.328 ± 4.563) was extremely significantly lower than in the females (197.636 ± 64.441) during the breeding season, (*p* < 0.01, Figure 2). 

In this study, the differences in urine T/E1G between the adult sexes during the breeding season were found (Figure 3), and the specific ratio is shown in Table 1. The ratio of the urine T/E1G of the male salamanders ranged from 174.675 to 367.470, and that of the females ranged from 0.238 to 0.606. The value of log_10_(T/E1G) of the adult male salamanders (2.429 ± 0.168) was extremely significantly higher than that of the adult females (−0.452 ± 0.210), (*p* < 0.01, Figure 3).

### 3.2. Verification of the Determination of CGSs’ Sex Using the T/E1G Ratio during the Non-Breeding Season

A total of 19 CGSs during the non-breeding season were prepared, including 7 adults and 12 sub-adults; the urine samples of each salamander were collected, and the salamanders were dissected to observe the gonads to determine the sex. The autopsy results showed that there were 3 males and 4 females among the 7 adult salamanders, and 7 males and 5 females among the 12 sub-adult salamanders. 

The hormone test results of the urine T from the 19 salamanders are shown in Table 2. The T concentration of the male salamanders (1175.511 ± 936.903) was extremely significantly higher than that of the females (149.370 ± 203.154) (*p* < 0.01, Figure 4). There were no significant differences between the adult and juvenile salamanders of the same sex. The results showed that the urine T concentration in the non-reproductive season was not related to adult status but was significantly related to sex difference.

The results of the urine hormone tests from the 19 CGSs during the non-breeding season are shown in Table 2. The urine E1G concentration varies greatly between individuals, with the male E1G concentrations (52.057 ± 58.885) lower than the female E1G concentrations (222.946 ± 218.721), and the difference was significant (*p* < 0.05). Similarly, there was no significant difference in E1G concentration between the adult and juvenile salamanders of the same sex (*p* > 0.05).

In our study, there was no regularity in the urine T or E1G in the male and female CGSs during the non-breeding season, except for the adult salamanders during the breeding season. Moreover, the value of urine log_10_(T/E1G) during the non-breeding season showed significant sexual differences (*p* < 0.05) (Figure 5 and Table 3). The value of the urine log_10_(T/E1G) of male CGSs was higher than 1, whereas that of the females was lower than 1. The accuracy of determining the sex of the adult CGSs during the breeding season was 100% when log_10_(T/E1G) was set to the limit of 1. The verification test of the log_10_(T/E1G) value “1” showed that all the 19 adult salamanders (both male and female salamanders) during the non-breeding season were completely consistent with the value “1”. That is, the urine log_10_(T/E1G) values of the male CGSs were all greater than 1 (between 1.050 and 2.490), and the values of the females were all less than 1 (between −2.155 and 0.720). The rate of accuracy for determining the sex of CGSs could reach 100% by using this value.

## 4. Discussion

Regarding sex differentiation, Klein and Bogart proposed a sex-determination hypothesis based on the ratio of androgen to estrogen in the gonads in the process of sexual differentiation. The hypothesis suggests that gonadal sex is determined by the local gonadal ratio of androgens to estrogen steroids. These androgens and estrogens compete to initiate and maintain different steroid-induced gene transcription pathways. In vertebrates, this steroid ratio is usually controlled by the activity of aromatase, which converts the androgen T into estrogen. The early ratio of androgen to estrogen can determine sex differentiation. When the ratio of androgen to estrogen is low, the ovary develops, whereas the testis develops when the ratio is high; this idea is known as the “equilibrium hypothesis” [34]. Our findings showed that T/E1G was lower in female CGSs and higher in male salamanders, which is in agreement with this “equilibrium hypothesis”. In this study, the urine T and E1G expression levels of the young CGSs (e.g., 2 years old and 3 years old) were lacking, and we will pay attention to this area in future studies.

Sex hormones are one of the main factors affecting sex determination in amphibian species [35]. The three common estrogens, estradiol, estrone, and estriol, can combine with the estrogen receptor to produce a range of biological effects. Studies have shown that estradiol from ovarian granulosa cells is rich in content and has strong activity, which can act on organs such as the uterus and the pituitary gland [35,36]. The primary metabolite is E1G, which was also commonly used to detect female ovarian function in basic research and clinical research [37]. At present, there are few studies on sex hormones in CGSs, and only the annual changes in the urine sex hormones and their relationship with gonadal development and reproduction have been reported [38]. 

However, to identify the sex of the animals, the ratio of estrogens to androgens in fecal samples was the most common measurement utilized [39,40]. No researchers had ever linked T/E1G ratio to the sex of animals; thus, it was difficult to discuss or compare the results of this study with those of other studies. The ability to measure sex hormone concentrations within individuals has important research and conservation implications. Conventional studies have used invasive blood collection to achieve this in the past [41,42,43,44]. Although collecting fecal samples is much less invasive and easier than collecting blood, there are also many disadvantages to this method [25]. In particular, for the aquatic animals such as the CGS, the feces are washed away or diluted when they defecate, and the urine is discharged directly into the water and cannot be collected directly. Indeed, we have not found the opportunity to sample the urine in other non-invasive approaches at present. The urine sampling method adopted in this study is easier to operate with minimal invasion than drawing blood from the tail vein. Compared with ultrasonography, our method is not limited only to adult CGSs during the breeding season but is also suitable for sub-adults, both in the breeding and non-breeding season. We hope to develop a urine detection kit for CGSs in the future, which will make the sex detection of CGSs more convenient, faster, and cheaper.

Both males and females produce androgens and estrogens due to the steroidogenic pathway (i.e., testosterone is a precursor of estradiol) [45,46]. The males of some species (e.g., pigs [47] and horses [48]) produce large amounts of estrogen compared with females of the same species. Although female neotropical otters (*Lontra longicaudis*) have higher levels of male hormones in fecal samples during the breeding season [40], in this study the urine T concentration of the male CGSs was significantly higher than that of the females, and the concentration of urine E1G of the males was less pronounced than that of the female salamanders. This differs from the *Lontra longicaudis* [40] and is consistent with the findings on bell frogs (*Litoria castanea*) [25].

Previous studies found that T and estradiol could be detected in the blood of both breeding and non-breeding CGSs, but there was no significant difference in the blood levels of the two hormones [32], which was different from the results of this study. This result may be caused by different tissues, or it may be related to the growth of CGSs, which needs further study.

As the breeding season of CGSs is during the flood season from June to July of every year, flood disasters affect the survival of artificially released CGSs. Consequently, the release of CGSs usually occurs in March to May before the flood season [49], which is during the non-breeding season of the salamanders. Therefore, the sex identification of the salamanders is particularly important at this time.

This is the first study to characterize the relationship between the T and E1G concentrations in the urine of CGSs. The findings should be further verified by increasing the sample size. In the future, this minimally invasive sampling method could be used to determine the sex of CGSs by measuring the urine sex hormones and also could provide a new approach for the sex identification of other vertebrates with no obvious secondary sex characteristics. We will increase the number of samples to verify the findings in this study in the future, especially CGS juveniles aged 2–3 years, during both the breeding and non-breeding seasons.

## 5. Conclusions

In conclusion, an interesting pattern of urine T and E1G was identified for CGSs. In other words, the sex of CGSs can be accurately determined by T/E1G in urine hormones; when there is a log_10_(T/E1G) value greater than 1, the salamander is male, and when the log_10_(T/E1G) value is less than 1, the salamander is female. This is applicable to both sub-adults and adults, as well as breeding and non-breeding salamanders.

## Figures and Tables

**Figure 1 animals-12-01112-f001:**
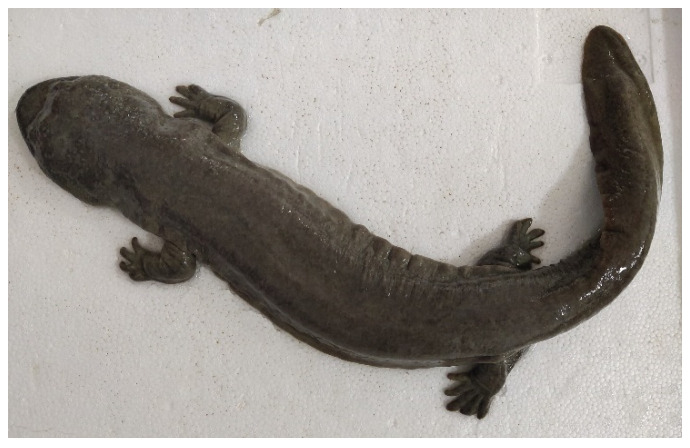
Photograph of Chinese giant salamander.

**Figure 2 animals-12-01112-f002:**
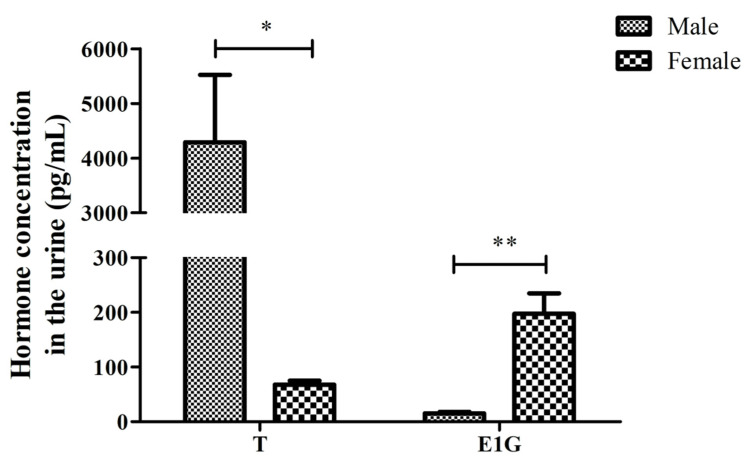
The T and E1G concentration in the urine of male and female adult CGSs during the breeding season (* *p* < 0.05, ** *p* < 0.01). The values are the means (±SD) of three salamanders per group.

**Figure 3 animals-12-01112-f003:**
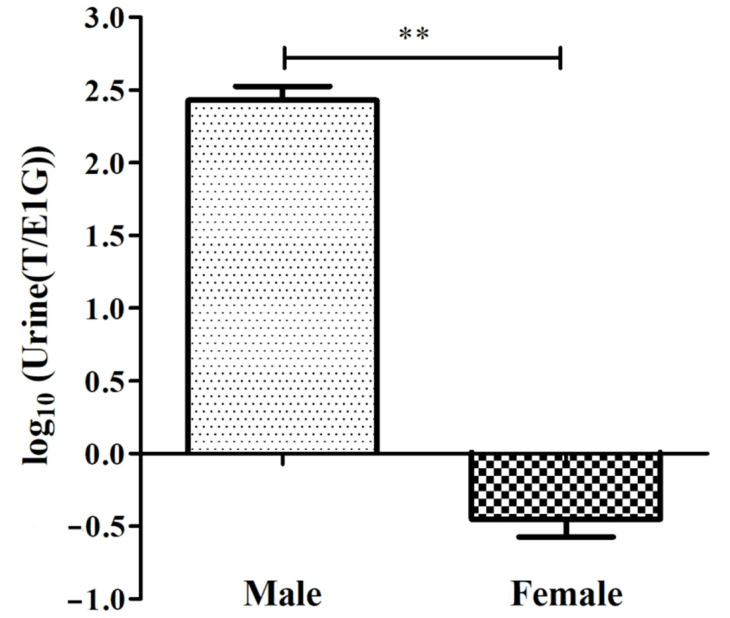
log_10_(Urine(T/E1G)) of male and female adult CGSs during the breeding season (** *p* < 0.01). The values are the means (±SD) of three salamanders per group.

**Figure 4 animals-12-01112-f004:**
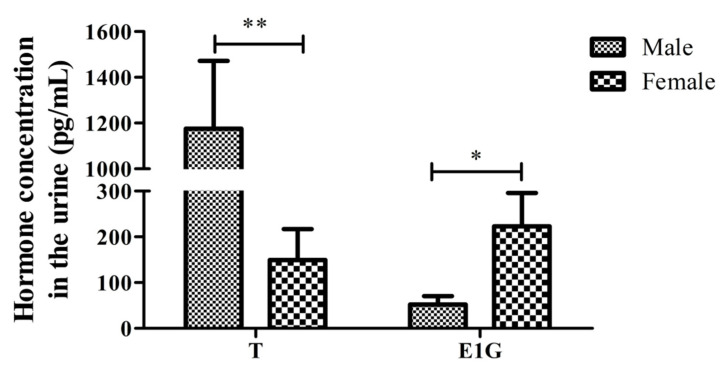
The T and E1G concentration in the urine of male and female CGSs during the non-breeding season (* *p* < 0.05, ** *p* < 0.01).

**Figure 5 animals-12-01112-f005:**
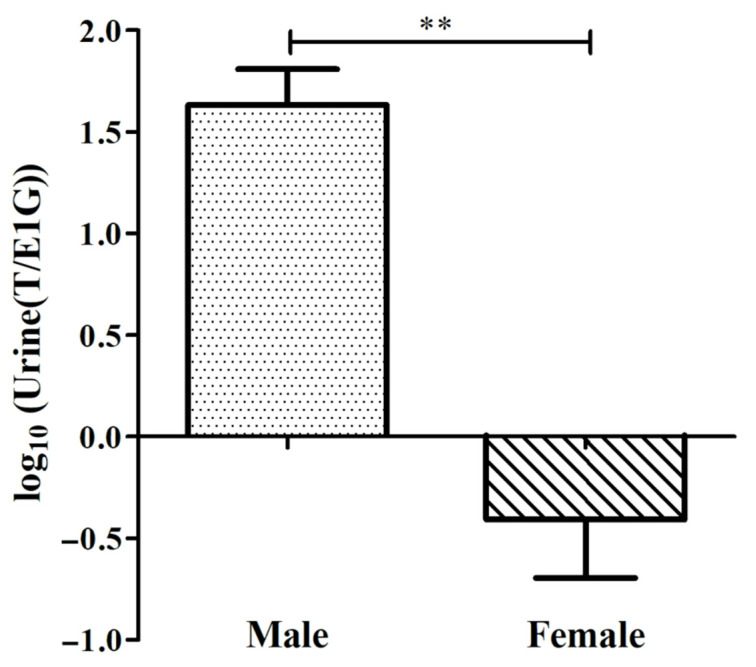
log_10_(Urine(T/E1G)) of male and female CGSs during the non-breeding season (** *p* < 0.01).

**Table 1 animals-12-01112-t001:** T and E1G concentration in the urine of adult CGSs during the breeding season.

Sex	No.	T		E1G		T/E1G	log_10_(T/E1G)
		OD	Conc. (pg/mL)	OD	Conc. (pg/mL)		
Male	M1	1.179	3052.5	1.580	10.1	302.736	2.481
	M2	1.178	3060.3	1.555	17.5	174.675	2.242
	M3	0.875	6754.5	1.552	18.4	367.470	2.565
Female	F1	2.151	82.1	1.159	135.5	0.606	−0.218
	F2	2.180	62.8	0.866	264.2	0.238	−0.623
	F3	2.186	59.0	1.011	193.2	0.306	−0.514

OD, Optical Density; conc., concentration.

**Table 2 animals-12-01112-t002:** The concentration of urine T and E1G of 19 CGSs during the non-breeding season.

Sex	No.	T (pg/mL)			E1G (pg/mL)		
		OD	Ind. Conc.	Aver. Conc.	OD	Ind. Conc.	Aver. Conc.
Male	SA1	2.217	138.6	1175.5 ± 936.9	1.608	0.8	52.1 ± 102.9
	SA2	2.089	438.6		1.576	38.2	
	SA3	1.986	790.6		1.607	2.6	
	SA4	2.146	285.7		1.607	1.5	
	SA5	1.792	1771.6		1.541	71.7	
	SA6	1.623	3066.4		1.560	54.1	
	SA7	1.832	1527.6		1.490	118.8	
	A1	1.750	2049.9		1.421	182.7	
	A2	2.083	454.9		1.585	28.9	
	A3	1.887	1231.2		1.592	21.3	
Female	SA8	2.275	51.7	149.4 ± 203.2	1.494	115.6	222.9 ± 218.7
	SA9	2.241	98.7		1.465	141.7	
	SA10	2.292	32.1		1.573	40.7	
	SA11	2.026	639.8		1.487	122.0	
	SA12	2.141	299.0		1.487	121.6	
	A4	2.266	63.0		1.305	294.2	
	A5	2.340	1.8		1.344	255.2	
	A6	2.280	45.2		0.930	769.6	
	A7	2.231	113.0		1.461	145.8	

SA, sub-adult CGSs; A, adult CGSs; OD, Optical Density; Ind. conc., Individual concentration; Aver. Conc., Average concentration.

**Table 3 animals-12-01112-t003:** The T/E1G ratio and log_10_(T/E1G) of 19 CGSs during the non-breeding season.

Sex	No.	T/E1G	log_10_(T/E1G)
Male	SA05	178.493	2.252
	SA06	11.477	1.060
	A07	309.311	2.490
	SA08	186.089	2.270
	SA09	24.707	1.393
	SA10	56.727	1.754
	SA12	12.856	1.109
	A01	11.223	1.050
	A02	15.734	1.197
	A03	57.703	1.761
Female	SA01	0.442	−0.355
	SA02	0.697	−0.157
	SA03	0.788	−0.103
	SA04	5.243	0.720
	SA11	2.460	0.391
	A04	0.214	−0.670
	A05	0.007	−2.155
	A06	0.059	−1.229
	A07	0.775	−0.111

Note: SA, sub-adult CGSs; A, adult CGSs.

## Data Availability

The data presented in this study are available on request from the corresponding author.

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
