# Peer review of "Using the Ratio of Urine Testosterone to Estrone-3-Glucuronide to Identify the Sex of Chinese Giant Salamanders (Andrias davidianus)"

_animals, 2022, doi:10.3390/ani12091112_

Round 1
Reviewer 1 Report
Giant Chinese salamanders are listed as Critically Endangered in the IUCN’s Red List, even though enormous numbers are bred each year in aquafarms. Using some of these salamanders to enhance dwindling natural populations is complicated by the need to ensure an equal sex balance in released individuals. Sex determination of adults is, however, difficult outside the breeding season and unreliable in juveniles. Hence the need for a reliable method of sex determination in this species. The authors have achieved this through examining the ratio of androgens to oestrogen in urine samples from salamanders of independently determined sex. Measurements of the ratio of androgens/oestrogens in plasma and faeces to detect the sex of an individual has been used widely in various mammals, reptiles and amphibians and is the obvious choice for this species. The methods for the determination of levels of testosterone (T) and oestrone-3-glucuronide (E1G), a metabolite of oestradiol, by enzymatic immunoassays in urine are carefully described and calibrated. A critical point that needs to be clarified in 2.1 in the MS is the independent way in which the sex of the tested animals was determined, as all of the verification relies on thus being correct. Standard curves for the two assays do not need to be included (Figs1 and 3) and the authors need to decide whether they wish to present their data in the form of tables (e.g. Table 1) or as figures (e.g. Fig 3). Both cannot be given as this is duplication of the data. I prefer using Figures as they are readily grasped by the reader. Also please note that it is ludicrous to report steroid concentrations to 3 decimal places! The EIAs are nowhere near that accurate, and even giving a single decimal place is sometimes misleading. The case that the authors have made for identifying sex based on the ratio of T:E1G is good and expressing the ratio as a logarithm simplifies differentiation.
Author Response
Point 1: A critical point that needs to be clarified in 2.1 in the MS is the independent way in which the sex of the tested animals was determined, as all of the verification relies on thus being correct.
Response 1: Dear Reviewer, thank you very much for your suggestion. The methods for determine the sex of the tested animals have been supplemented in 2.2 in the revised manuscript.
Point 2: Standard curves for the two assays do not need to be included (Figs1 and 3) and the authors need to decide whether they wish to present their data in the form of tables (e.g. Table 1) or as figures (e.g. Fig 3). Both cannot be given as this is duplication of the data. I prefer using Figures as they are readily grasped by the reader.
Response 2: Dear Reviewer, thank you for your suggestion. After considering your comments and suggestions from another reviewer, we have deleted the Figure 1 and Figure 3 in the MS.
Point 3: Also please note that it is ludicrous to report steroid concentrations to 3 decimal places! The EIAs are nowhere near that accurate, and even giving a single decimal place is sometimes misleading. The case that the authors have made for identifying sex based on the ratio of T:E1G is good and expressing the ratio as a logarithm simplifies differentiation.
Response 3: Dear Reviewer, thank you very much for your suggestion, the concentrations of T and E1G have been modified and left with 1 decimal places.
Reviewer 2 Report
This seems like an important manuscript concerning a very endangered species. I think these results will have important uses in the recovery of the species. Since this is an unusual species found only in one country, I think it would be useful to show a photograph of the Chinese giant salamander in the introduction. From my google search it is quite amazing.
I would not recommend using a question for the title. While there is nothing wrong with a question I would prefer:
"Using the ratio of urine testosterone to Estrone-3-glucuronide to identify the sex of Chinese giant salamanders (Andrias davidianus)."
The text seems wordy. One suggestion would be, in place of using the term "Chinese giant salamander" every time, the authors could use an abbreviation such as (Cgs) occasionally. Thus, the first use of Chinese giant salamander (Cgs) and after that use Cgs to save space?
Other comments:
Abstract line 7 : should not say "extremely significantly higher". Could say "and even more pronounced during the non-breeding season (p < 0.01)."
Generally we do not say " (p<0.05) and extremely significant difference (p < 0.01)". It is either significant or not significant. So in this case one would say significant at the (p < 0.05) level or significant at the (p < 0.01) level.
Generally the manuscript is well written with clear references and accurate analyses and figures.
The use of the urine log10(T/E1G) is a very good option with clear results.
The use of urine for obtaining the samples is actually a very old method but very useful in this situation with this unusual species.
Author Response
Comments and Suggestions for Authors :
Point 1: This seems like an important manuscript concerning a very endangered species. I think these results will have important uses in the recovery of the species. Since this is an unusual species found only in one country, I think it would be useful to show a photograph of the Chinese giant salamander in the introduction. From my google search it is quite amazing.
Response 1: Dear Reviewer, we authors agree with your suggestion, we have shown a photograph of the Chinese giant salamander in the introduction.
Point 2: I would not recommend using a question for the title. While there is nothing wrong with a question I would prefer: "Using the ratio of urine testosterone to Estrone-3-glucuronide to identify the sex of Chinese giant salamanders (Andrias davidianus)."
Response 2: Dear Reviewer, we authors agree with your suggestion, and we changed the title according to your suggestion.
Point 3: The text seems wordy. One suggestion would be, in place of using the term "Chinese giant salamander" every time, the authors could use an abbreviation such as (Cgs) occasionally. Thus, the first use of Chinese giant salamander (Cgs) and after that use Cgs to save space?
Response 3: Dear Reviewer, we authors agree with your suggestion, the abbreviation CGS for Chinese giant salamander and CGSs for Chinese giant salamanders had be used in the manuscript, respectively.
Other comments:
Abstract line 7: should not say "extremely significantly higher". Could say "and even more pronounced during the non-breeding season (p < 0.01)." Generally we do not say " (p<0.05) and extremely significant difference (p < 0.01)". It is either significant or not significant. So in this case one would say significant at the (p < 0.05) level or significant at the (p < 0.01) level.
Response 4: Dear Reviewer, we authors agree with your suggestion and have revised it accordingly.
Reviewer 3 Report
I have read the the paper entitle: "Can the ratio of urine testosterone to Estrone-3-glucuronide identify the sex of Chinese giant salamanders (Andrias davidianus)?"
The aim is very intersting and useful but the presentation of results need to be improved. Please check and rephrase all material and methods, in particular:
- add ethics approval for animal studies
- add info regarding the husbndry and managememnt of the animals
- sexing of animals
- statistical analysis: to report the nomality test used.
Result: please rephrase this part to a more clear presentation of data:
- to move standard curves on supplementary materials
- report all data of animals for T and E1G in the same image/table
Minor suggestions:
- line 88: explain why ultrasonography is an no invasive method.
Author Response
Point 1: add ethics approval for animal studies
Response 1: Dear Reviewer, thank you very much for your suggestion and we have added relevant content.
Point 2: add info regarding the husbndry and managememnt of the animals
Response 2: Dear Reviewer, we authors agree with your suggestion and the info had been added in 2.2 in the revised manuscript.
Point 3: sexing of animals
Response 3: Dear Reviewer, The methods for sexing of animals have been supplemented in 2.2 in the revised manuscript.
Point 4: statistical analysis: to report the nomality test used.
Response 4: Dear Reviewer, The nomality test used have been supplemented in 2.6.
Point 5: Result: please rephrase this part to a more clear presentation of data: to move standard curves on supplementary materials
Response 5: Dear Reviewer, thank you for your suggestion. After considering your comments and suggestions from another reviewer, we have rephrased this part to a more clear presentation of data, and deleted the Figures of standard curve in the manuscript.
Point 6: report all data of animals for T and E1G in the same image/table
Response 6: Dear Reviewer, we authors agree with your suggestion and we have mad the change.
Minor suggestions:
Point 7: - line 88: explain why ultrasonography is an no invasive method.
Response 7: Dear Reviewer, what we were trying to say in the manuscript is that ultrasonography is more expensive and only suitable for adult CGS during breeding season, and draw blood is invasive to CGSs The previous expression is not clear, we have made modifications. In fact, ultrasonography is also a minimally invasive method.
Round 2
Reviewer 3 Report
Dear authors,
I have read the revised version. I suggest to add info regarding the methods of sampling of the urine: methods of euthanisia, needle used...
In the discussion add info and consideration regarding the type of sampling compared to the others (please to refere also at the introduction considering the blood sampling and the ultrasonography), the invasive vs no invasive methods and the opportunity to sample the urine in other methods (no invasive), ecc.
Author Response
point 1: I suggest to add info regarding the methods of sampling of the urine: methods of euthanisia, needle used...
Dear Reviewer, thank you very much for your suggestion, We added the relevant information in lines 123 to 128 of manuscript, At the same time, a reference [33] published by our laboratory is added. The specific additions are as follows:
The salamanders were deep anesthetized in a water bath containing tricaine methane sulfonate (MS-222) at a concentration of 600 mg·L-1. The method of anesthesia was based on the results of our previous study [33], and the anesthesia lasted about 35 minutes for 7.5 years old CGS and 30 minutes for 4.5 years old CGS. Finally, the deep anesthetized CGSs were euthanized for dissection to confirm their sex after the urine was collected.
point 2: In the discussion add info and consideration regarding the type of sampling compared to the others (please to refere also at the introduction considering the blood sampling and the ultrasonography), the invasive vs no invasive methods and the opportunity to sample the urine in other methods (no invasive), ecc.
Dear Reviewer, thank you very much for your suggestion. We added the relevant information in lines 272 to 281 of manuscript. The specific additions are as follows:
In particular, for the aquatic animals such as CGS, the feces are washed away or diluted when they defecate, the urine is discharged directly into the water and cannot be collected directly. Indeed, we have not find the opportunity to sample the urine in other no invasive approach at present. The urine sampling method adopted in this study is easier to operate with minimal invasion than draw blood from tail vein. Compared with ultrasonography, our method is not only limited to adult CGS during breeding season, but also suitable for sub-adults, both in breeding and non-breeding season. We hope to develop a urine detection kit for CGS in the future, which will make sex detection of CGS more convenient, faster and cheaper.
